# Learning Loop Invariants for Program Verification

**Xujie Si**[*]
University of Pennsylvania
xsi@cis.upenn.edu

**Hanjun Dai** [*]
Georgia Tech
hanjundai@gatech.edu

**Mukund Raghothaman**
University of Pennsylvania
rmukund@cis.upenn.edu

**Mayur Naik**
University of Pennsylvania
mhnaik@cis.upenn.edu

**Le Song**
Georgia Tech and Ant Financial
lsong@cc.gatech.edu

## Abstract

A fundamental problem in program verification concerns inferring loop invariants. The problem is undecidable and even practical instances are challenging. Inspired by how human experts construct loop invariants, we propose a reasoning framework CODE2INV that constructs the solution by multi-step decision making and querying an external program graph memory block. By training with reinforcement learning, CODE2INV captures rich program features and avoids the need for ground truth solutions as supervision. Compared to previous learning tasks in domains with graph-structured data, it addresses unique challenges, such as a binary objective function and an extremely sparse reward that is given by an automated theorem prover only after the complete loop invariant is proposed. We evaluate CODE2INV on a suite of 133 benchmark problems and compare it to three state-of-the-art systems. It solves 106 problems compared to 73 by a stochastic search-based system, 77 by a heuristic search-based system, and 100 by a decision tree learning-based system. Moreover, the strategy learned can be generalized to new programs: compared to solving new instances from scratch, the pre-trained agent is more sample efficient in finding solutions.

## 1 Introduction

The growing ubiquity and complexity of software has led to a dramatic increase in software bugs and security vulnerabilities that pose enormous costs and risks. Program verification technology enables programmers to prove the absence of such problems at compile-time before deploying their program. One of the main activities underlying this technology involves inferring a *loop invariant*—a logical formula that constitutes an abstract specification of a loop—for each loop in the program. Obtaining loop invariants enables a broad and deep range of correctness and security properties to be proven automatically by a variety of program verification tools spanning type checkers, static analyzers, and theorem provers. Notable examples include Microsoft Code Contracts for .NET programs [1] and the Verified Software Toolchain spanning C source code to machine language [2].

Many different approaches have been proposed in the literature to infer loop invariants. The problem is undecidable, however, and even practical instances are challenging, which greatly limits the benefits of program verification technology. Existing approaches suffer from key drawbacks: they are purely search-based, or they use hand-crafted features, or they are based on supervised learning. The performance of search-based approaches is greatly hindered by their inability to learn from past mistakes. Hand-crafted features limit the space of possible invariants, e.g., Garg et al. [3] is limited to features of the form $x \pm y \le c$ where $c$ is a constant, and thus cannot handle invariants that involve $x + y \le z$ for program variables $x, y, z$. Finally, obtaining ground truth solutions needed by supervised learning is hindered by the undecidability of the loop invariant generation problem.

In this paper, we propose CODE2INV, an end-to-end learning-based approach to infer loop invariants. CODE2INV has the ability to automatically learn rich latent representations of desirable invariants,

---

[*]Both authors contributed equally to the paper.

and can avoid repeating similar mistakes. Furthermore, it leverages reinforcement learning to discover invariants by partial feedback from trial-and-error, without needing ground truth solutions for training.

The design of CODE2INV is inspired by the reasoning exercised by human experts. Given a program, a human expert first maps the program to a well-organized structural representation, and then composes the loop invariant step by step. Based on such reasoning, different parts of the representation get highlighted at each step. To mimic this procedure, we utilize a graph neural network model (GNN) to construct the structural external memory representation of the program. The multi-step decision making is implemented by an autoregressive model, which queries the external memory using an attention mechanism. The decision at each step is a syntax- and semantics-guided decoder which generates subparts of the loop invariant.

CODE2INV employs a reinforcement learning approach since it is computationally intensive to obtain ground truth solutions. Although reinforcement learning algorithms have shown remarkable success in domains like combinatorial optimization [4, 5] (see Section 6 for more discussion on related work), our setting differs in two crucial ways: first, it has a non-continuous objective function (i.e., a proposed loop invariant is correct or not); and second, the positive reward is extremely sparse and given only after the correct loop invariant is proposed, by an automated theorem prover [6]. We therefore model the policy learning as a multi-step decision making process: it provides a fine-grained reward at each step of building the loop invariant, followed by continuous feedback in the last step based on counterexamples collected by the agent itself during trial-and-error learning.

We evaluate CODE2INV on a suite of 133 benchmark problems from recent works [3, 7, 8] and the 2017 SyGuS program synthesis competition [9]. We also compare it to three state-of-the-art systems: a stochastic search-based system C2I [10], a heuristic search-based system LOOPINVGEN [8], and and a decision tree learning-based system ICE-DT [3]. CODE2INV solves 106 problems, versus 73 by C2I, 77 by LOOPINVGEN, and 100 by ICE-DT. Moreover, CODE2INV exhibits better learning, making orders-of-magnitude fewer calls to the theorem prover than these systems.

## 2  Background

We formally define the loop invariant inference and learning problems by introducing Hoare logic [11], which comprises a set of axioms and inference rules for proving program correctness assertions. Let $P$ and $Q$ denote predicates over program variables and let $S$ denote a program. We say that *Hoare triple* $\{P\}\,S\,\{Q\}$ is valid if whenever $S$ begins executing in a state that satisfies $P$ and finishes executing, then the resulting state satisfies $Q$. We call $P$ and $Q$ the *pre-condition* and *post-condition* respectively of $S$. Hoare rules allow to derive such triples inductively over the structure of $S$. The rule most relevant for our purpose is that for loops:

$$\frac{P \Rightarrow I \quad (\textit{pre}) \qquad \{I \wedge B\}\,S\,\{I\} \quad (\textit{inv}) \qquad (I \wedge \neg B) \Rightarrow Q \quad (\textit{post})}{\{P\}\ \mathbf{while}\ B\ \mathbf{do}\ S\ \{Q\}}$$

Predicate $I$ is called a *loop invariant*, an assertion that holds before and after each iteration, as shown in the premise of the rule. We can now formally state the loop invariant inference problem:

**Problem 1 (Loop Invariant Inference):** Given a pre-condition $P$, a post-condition $Q$ and a program $S$ containing a single loop, can we find a predicate $I$ such that $\{P\}\,S\,\{Q\}$ is valid?

Given a candidate loop invariant, it is straightforward for an automated theorem prover such as Z3 [6] to check whether the three conditions denoted *pre*, *inv*, and *post* in the premise of the above rule hold, and thereby prove the property asserted in the conclusion of the rule. If any of the three conditions fails to hold, the theorem prover returns a concrete counterexample witnessing the failure.

The loop invariant inference problem is undecidable. Moreover, even seemingly simple instances are challenging, as we illustrate next using the program in Figure 1(a). The goal is to prove that assertion $(y > 0)$ holds at the end of the program, for every input value of integer variable $y$. In this case, the pre-condition $P$ is true since the input value of $y$ is unconstrained, and the post-condition $Q$ is $(y > 0)$, the assertion to be proven. Using predicate $(x < 0 \lor y > 0)$ as the loop invariant $I$ suffices to prove the assertion, as shown in Figure 1(b). Notation $\phi[e/x]$ denotes the predicate $\phi$ with each occurrence of variable $x$ replaced by expression $e$. This loop invariant is non-trivial to infer. The reasoning is simple in the case when the input value of $y$ is non-negative, but far more subtle in the case when it is negative: regardless of how negative it is at the beginning, the loop will iterate at least as many times as to make it positive, thereby ensuring the desired assertion upon finishing. Indeed, a state-of-the-art loop invariant generator LOOPINVGEN [8] crashes on this problem instance after making 1,119 calls to Z3, whereas CODE2INV successfully generates it after only 26 such calls.

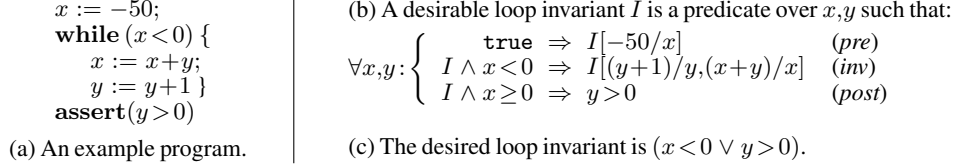

$$x := -50;$$
$$\textbf{while } (x<0) \{$$
$$\quad x := x+y;$$
$$\quad y := y+1 \}$$
$$\textbf{assert}(y>0)$$

(a) An example program.

(b) A desirable loop invariant $I$ is a predicate over $x,y$ such that:

$$\forall x,y : \begin{cases} \texttt{true} & \Rightarrow & I[-50/x] & \textit{(pre)} \\ I \wedge x<0 & \Rightarrow & I[(y+1)/y,(x+y)/x] & \textit{(inv)} \\ I \wedge x\geq 0 & \Rightarrow & y>0 & \textit{(post)} \end{cases}$$

(c) The desired loop invariant is $(x<0 \vee y>0)$.

Figure 1: A program with a correctness assertion and a loop invariant that suffices to prove it.

The central role played by loop invariants in program verification has led to a large body of work to automatically infer them. Many previous approaches are based on exhaustive bounded search using domain-specific heuristics and are thereby limited in applicability and scalability [7, 12–18]. A different strategy is followed by data-driven approaches proposed in recent years [3, 8, 10]. These methods speculatively guess likely invariants from program executions and check their validity. In [3], decision trees are used to learn loop invariants with simple linear features, e.g. $a*x+b*y<c$, where $a,b\in\{-1,0,1\}, c\in Z$. In [8], these features are generalized by systematic enumeration. In [10], stochastic search is performed over a set of constraint templates. While such features or templates perform well in specific domains, however, they may fail to adapt to new domains. Moreover, even in the same domain, they do not benefit from past experiences: successfully inferring the loop invariant for one program does not speed up the process for other similar ones. We hereby formulate the second problem we aim to address:

**Problem 2 (Loop Invariant Learning):** Given a set of programs $\{S_i\} \sim \mathcal{P}$ that are sampled from some unknown distribution $\mathcal{P}$, can we learn from them and generalize the strategy we learned to other programs $\{\tilde{S}_i\}$ that are from the same distribution?

## 3 End-to-End Reasoning Framework

### 3.1 The reasoning process of a human expert

We start out by illustrating how a human expert might typically accomplish the task of inferring a loop invariant. Consider the example in Figure 2 chosen from our benchmarks.

An expert usually starts by reading the assertion (line 15), which contains variables x and y, then determines the locations where these two variables are initialized, and then focuses on the locations where they are updated in the loop. Instead of reasoning about the entire assertion at once, an expert is likely to focus on updates to one variable at a time. This reasoning yields the observation that x is initialized to zero (line 2) and may get incremented in each iteration (line 5,9). Thus, the sub goal "x < 4" may not always hold, given that the loop iterates non-deterministically. This in turn forces the other part "y > 2" to be true when "x >= 4". The only way x can equal or exceed 4 is to execute the first if branch 4 times (line 4-6), during which y is set to 100. Now, a natural guess for the loop invariant is "x < 4 || y >= 100". The reason for guessing "y >= 100" instead of "y <= 100" is because part of the proof goal is "y > 2".

```
1   int main() {
2     int x = 0, y = 0;
3     while (*) {
4       if (*) {
5         x++;
6         y = 100;
7       } else if (*) {
8         if (x >= 4) {
9           x++;
10          y++;
11        }
12        if (x < 0) y--;
13      }
14    }
15    assert( x < 4 || y > 2);
16  }
```

Figure 2: An example from our benchmarks. "*" denotes non-deterministic choice.

However, this guess will be rejected by the theorem prover. This is because y might be decreased by an arbitrary number of times in the third if-branch (line 12), which happens when x is less than zero; to avoid that situation, "x >= 0" should also be part of the loop invariant. Finally, we have the correct loop invariant: "(x >= 0) && (x < 4 || y >= 100)", which suffices to prove the assertion.

We observe that the entire reasoning process consists of three key components: 1) organize the program in a hierarchical-structured way rather than a sequence of tokens; 2) compose the loop invariant step by step; and 3) focus on a different part of the program at each step, depending on the inference logic, e.g., abduction and induction.

### 3.2 Programming the reasoning procedure with neural networks

We propose to use a neural network to mimic the reasoning used by human experts as described above. The key idea is to replace the above three components with corresponding differentiable modules:

- a structured external memory representation which encodes the program;
- a multi-step autoregressive model for incremental loop invariant construction; and
- an attention component that mimics the varying focus in each step.

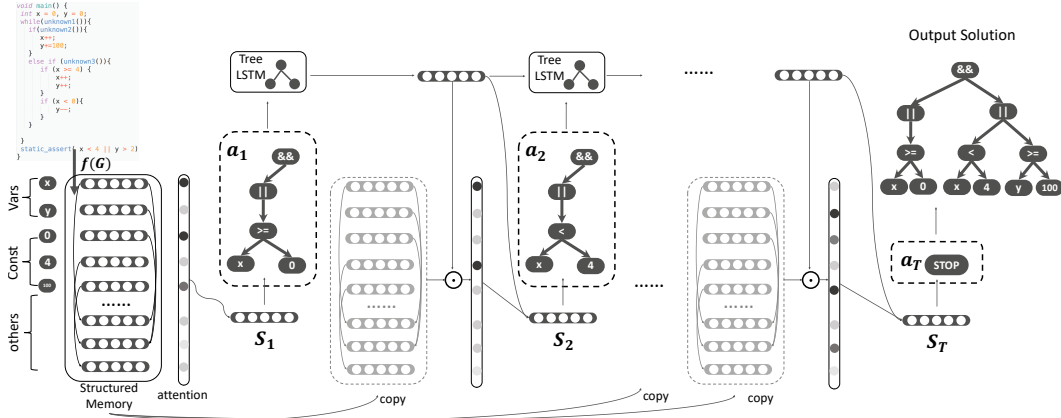

Figure 3: Overall framework of neuralizing loop invariant inference.

As shown in Figure 3, these modules together build up the network that constructs loop invariants from programs, while being jointly trained with reinforcement learning described in Section 4. At each step, the neural network generates a predicate. Then, given the current generated partial tree, a TreeLSTM module summarizes what have been generated so far, and the summarization is used to read the memory using attention. Lastly, the summarization together with the read memory is fed into next time step. We next elaborate upon each of these three components.

### 3.2.1 Structured external memory

The loop invariant is built within the given context of program. Thus it is natural to encode the program as an external memory module. However, in contrast to traditional memory networks [19, 20], where the memory slots are organized as a linear array, the information contained in a program has rich structure. A chain LSTM over program tokens can in principle capture such information but it is challenging for neural networks to understand with limited data. Inspired by Allamanis et al. [21], we instead use a graph-structured memory representation. Such a representation allows to capture rich semantic knowledge about the program such as its control-flow and data-flow.

More concretely, we first convert a given program into static single assignment (SSA) form [22], and construct a control flow graph, each of whose nodes represents a single program statement. We then transform each node into an abstract syntax tree (AST) representing the corresponding statement. Thus a program can be represented by a graph $G = (V, E)$, where $V$ contains terminals and nonterminals of the ASTs, and $E = \{(e_x^{(i)}, e_y^{(i)}, e_t^{(i)})\}_{i=1}^{|E|}$ is the set of edges. The directed edge $(e_x^{(i)}, e_y^{(i)}, e_t^{(i)})$ starts from node $e_x^{(i)}$ to $e_y^{(i)}$, with $e_t^{(i)} \in \{1, 2, ..., K\}$ representing edge type. In our construction, the program graph contains 3 different edge types (and 6 after adding reversed edges).

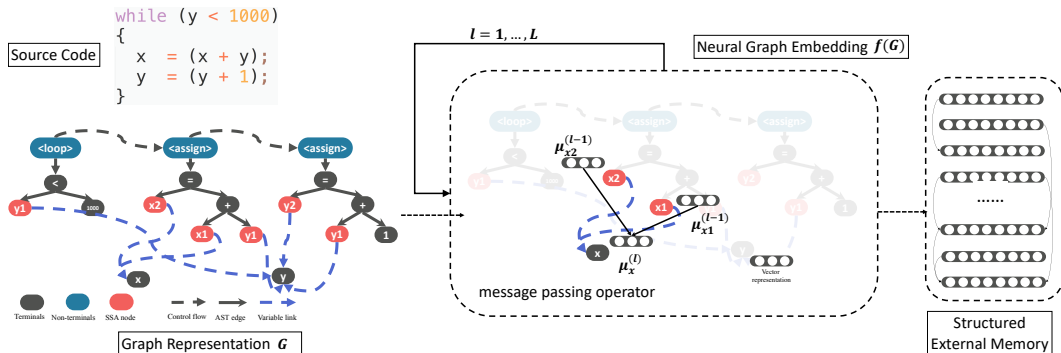

Figure 4: Diagram for source code graph as external structured memory. We convert a given program into a graph $G$, where nodes correspond to syntax elements, and edges indicate the control flow, syntax tree structure, or variable linking. We use embedding neural network to get structured memory $f(G)$.

To convert the graph into vector representation, we follow the general message passing operator introduced in graph neural network (GNN) [23] and its variants [21, 24, 25]. Specifically, the graph

network will associate each node $v \in V$ with an embedding vector $\mu_v \in \mathbb{R}^d$. The embedding is updated iteratively using the general neighborhood embedding as follows:

$$\mu_v^{(l+1)} = h(\{\mu_u^{(l)}\}_{u \in \mathcal{N}^k(v), k \in \{1,2,...,K\}}) \tag{1}$$

Here $h(\cdot)$ is a nonlinear function that aggregates the neighborhood information to update the embedding. $\mathcal{N}^k(v)$ is the set of neighbor nodes connected to $v$ with edge type $k$, *i.e.*, $\mathcal{N}^k(v) = \{u|(u,v,k) \in E\}$. Such process will be repeated for $L$ steps, and the node embedding $\mu_v$ is set to $\mu_v^{(L)}, \forall v \in V$. Our parameterization takes the edge types into account. The specific parameterization used is shown below:

$$\mu_v^{(l+1),k} = \sigma(\sum_{u \in \mathcal{N}^k(v)} \mathbf{W}_2 \mu_u^{(l)}), \forall k \in \{1,2,...,K\} \tag{2}$$

$$\mu_v^{(l+1)} = \sigma(\mathbf{W}_3[\mu_v^{(l+1),1}, \mu_v^{(l+1),2}, ..., \mu_v^{(l+1),K}]) \tag{3}$$

with the boundary case $\mu_v^{(0)} = \mathbf{W}_1 \mathbf{x}_v$. Here $\mathbf{x}_v$ represents the syntax information of node $v$, such as token or constant value in the program. Matrices $\mathbf{W}_{1,2,3}$ are learnable model parameters, and $\sigma$ is some nonlinear activation function. Figure 4 shows the construction of graph structured memory using iterative message passing operator in Eq (1). $f(G) = \{\mu_v\}_{v \in V}$ denotes the structured memory.

### 3.2.2 Multi-step decision making process

A loop invariant itself is a mini-program that contains expressions and logical operations. Without loss of generality, we define the loop invariant to be a tree $\mathcal{T}$, in a form with conjunctions of disjunctions:

$$\mathcal{T} = (\mathcal{T}_1 \,||\, \mathcal{T}_2 ...) \,\&\& \,(\mathcal{T}_{t+1} \,||\, \mathcal{T}_{t+2}...) \,\&\&\, ... \,(...\mathcal{T}_{T-1} \,||\, \mathcal{T}_T) \tag{4}$$

Each subtree $\mathcal{T}_t$ is a simple logic expression (*i.e.*, x < y * 2 + 10 - z). Given this representation form, it is natural to use Markov decision process (MDP) to model this problem, where the corresponding $T$-step finite horizon MDP is defined as $\mathcal{M}^G = (s_1, a_1, r_1, s_2, a_2, ..., s_T)$. Here $s_t, a_t, r_t$ represent the state, action and reward at time step $t = 1, ..., T-1$, respectively. Here we describe the state and action used in the inference model, and describe the design of reward and termination in Section 4.

**action:** As defined in Eq (4), a loop invariant tree $\mathcal{T}$ consists of multiple subtrees $\{\mathcal{T}_t\}$. Thus we model the action at time step $t$ as $a_t = (op_t, \mathcal{T}_t)$, where $op_t$ can either be $||$ or $\&\&$. That is to say, at each time step, the agent first decides whether to attach the subexpression $\mathcal{T}_t$ to an existing disjunction, or create a new disjunction and add it to the list of conjunctions. We use $\mathcal{T}^{(<t)}$ to denote the partial tree generated by time $t$ so far. So the policy $\pi(\mathcal{T}|G)$ is decomposed into:

$$\pi(\mathcal{T}|G) = \prod_{t=1}^{T} \pi(a_t|\mathcal{T}^{(<t)}, G) = \prod_{t=1}^{T} \pi(op_t, \mathcal{T}_t|\mathcal{T}^{(<t)}, G) \tag{5}$$

where $\mathcal{T}^{(<1)}$ is empty at the first step. The generation process of subtree $\mathcal{T}_t$ is also an autoregressive model implemented by LSTM. However, generating a valid program is nontrivial, since strong syntax and semantics constraints should be enforced. Recent advances in neural program synthesis [26, 27] utilize formal language information to help the generation process. Here we use the Syntax-Directed decoder proposed in [28] to guarantee both the syntax and semantics validity. Specifically,

- Syntax constraints: The AST generation follows the grammar of loop invariants described in Eq 4. Operators such as +, -, * are non-terminal nodes in the AST while operands such as constants or variables are leaf nodes.
- Semantic constraints: We regulate the generated loop invariant to be meaningful. For example, a valid loop invariant must contains all the variables that appear in the given assertion. Otherwise, the missing variables can take arbitrary values, causing the assertion to be violated. In contrast to offline checking which discards invalid programs after generation, such online regulation restricts the output space of the program generative model, which in turn makes learning efficient.

**state:** At time step $t = 1$, the state is simply the weighted average of structured memory $f(G)$. At each later time step $t > 1$, the action $a_t$ should be conditioned on graph memory, as well as the partial tree generated so far. Thus $s_t = (G, \mathcal{T}^{(<t)})$.

### 3.2.3 Memory query with attention

At different time steps of inference, a human usually focuses on different parts of program. Thus the attention mechanism is a good choice to mimic such process. Specifically, at time step $t$, to summarize what we have generated so far, we use TreeLSTM [29] to embed the partial tree $\mathcal{T}^{(<t)}$. Then the

embedding of partial tree $\mathbf{v}_{\mathcal{T}^{(<t)}} = \text{TreeLSTM}(\mathcal{T}^{(<t)})$ is used as the query to read the structured memory $f(G)$. Specifically, $\text{read}(f(G), \mathbf{v}_{\mathcal{T}^{(<t)}}) = \sum_{v \in \mathcal{V}} \alpha_v \mu_v$ and $\alpha_v = \frac{\exp \mu_v^\top \mathbf{v}_{\mathcal{T}^{(<t)}}}{\sum_{v \in V} \exp \mu_v^\top \mathbf{v}_{\mathcal{T}^{(<t)}}}$ are the corresponding attention weights.

# 4 Learning

The undecidability of the loop invariant generation problem hinders the ability to obtain ground truth solutions as supervisions for training. Inspired by recent advances in combinatorial optimization [4, 5], where the agent learns a good policy by trial-and-error, we employ reinforcement learning to learn to propose loop invariants. Ideally, we seek to learn a policy $\pi(\mathcal{T}|G)$ that proposes a correct loop invariant $\mathcal{T}$ for a program graph $G$. However, directly solving such a model is practically not feasible, since:

- In contrast to problems tackled by existing work, where the objective function is relatively continuous (*e.g.*, tour length of traveling salesman problem), the proposed loop invariant only has binary objective (*i.e.*, correct or not). This makes the loss surface of the objective function highly non-smooth.
- Finding the loop invariant is a bandit problem where the binary reward is given only after the invariant is proposed. Also, in contrast to two player games [30] where a default policy (e.g., random rollout) can be used to estimate the reward, it is a single player game with an extremely sparse reward.

To tackle the above two challenges, the multi-step decision making model proposed in Section 3.2.2 is used, where a fine-grained reward is also designed for each step. In the last step, a continuous feedback is provided based on the counterexamples collected by the agent itself.

## 4.1 Reinforcement learning setup

Section 3.2.2 defines the state and action representation used for inference. We next describe our setup of the environment which is important to properly train a reinforcement learning agent.

**reward:** In each intermediate step $t \in 1,...,T-1$, an intermediate reward $r_t$ is given to regulate the generation process. For example, a subexpression should be non-trivial, and it should not contradict $\mathcal{T}^{(<t)}$. In the last step, the generated loop invariant $\mathcal{T}$ is given to a theorem prover, which returns success or failure. In the latter case, the theorem prover also tells which step (*pre*, *inv*, *post*) failed, and provides a counterexample. The failure step can be viewed as a "milestone" of the verification process, providing a coarse granularity feedback. To achieve continuous (i.e. fine granularity) reward within each step, we exploit the counterexamples collected so far. For instance, the ratio of passed examples is a good indicator of the learning progress. Appendix **??** describes the reward design in detail.

**termination:** There are several conditions that may trigger the termination of tree generation: (1) the agent executes the "stop" action, as illustrated in Figure 3; (2) the generated tree has the maximum number of branches allowed; or (3) the agent generates an invalid action.

## 4.2 Training of the learning agent

We use advantage actor critic (A2C) to train the above reinforcement learning policy. Specifically, let $\theta = \{\mathbf{W}_i\}$ be the parameters in graph memory representation $f(\cdot; \theta)$, and $\phi$ be the parameter used in $\pi(a_t|\mathcal{T}^{(<t)}, G; \phi)$, our objective is to maximize the expected policy reward:

$$\max_{\theta,\phi} \mathbb{E}_{\pi(op_t, \mathcal{T}_t | \mathcal{T}^{(<t)}, G; \phi)} \left( \sum_{t'=t}^{T} \gamma^{t'-t} r_{t'} - b(\mathcal{T}^{(<t)}, G; \psi) \right) \qquad (6)$$

The baseline function $b(\mathcal{T}^{(<t)}, G; \psi)$ parameterized by $\psi$ is used to estimate the expected return, so as to reduce the variance of policy gradient. It is trained to minimize $\mathbb{E}_{\pi,t} \| \sum_{t'=t}^{T} \gamma^{t'-t} r_{t'} - b(\mathcal{T}^{(<t)}, G; \psi) \|$. We simply apply two layer fully connected neural network to predict the expected return. $\gamma$ is the discounting factor. Since the MDP is finite horizon, we use $\gamma = 1$ to address the long-term reward.

# 5 Experiments

We evaluate CODE2INV on a suite of 133 benchmark programs from recent works [3, 7, 8] and the 2017 SyGuS competition [31].[2] Each program consists of three parts: a number of assumption or assignment statements, one loop which contains nested if-else statements with arithmetic operations, and one assertion statement. Appendix **??** provides more details about the dataset and the competition.

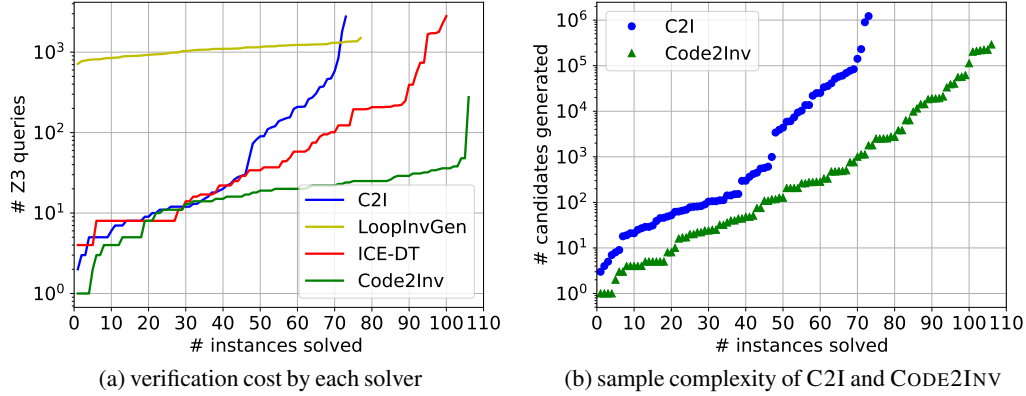

Figure 5: Comparison of CODE2INV with state-of-the-art solvers on benchmark dataset.

We first evaluate CODE2INV as an out-of-the-box solver, i.e., without any training or fine-tuning with respect to the dataset. We then conduct an ablation study to justify various design choices. Finally, we evaluate the impact of training CODE2INV on a similar dataset.

## 5.1 Finding loop invariants from scratch

In this section, we study the capability of CODE2INV with no training, that is, using it as an out-of-the-box solver. We compare CODE2INV with three state-of-the-art solvers: C2I [10], which is based on stochastic search; LOOPINVGEN [8], which searches a conjunctive normal form over predicates synthesized by an underlying engine, ESCHER [17]; and ICE-DT [3], which learns a decision tree over manually designed features (e.g. predicate templates). The last two solvers are the winners of the invariant synthesis track of the SyGuS 2017 and 2016 competitions, respectively.

A uniform metric is needed to compare the different solvers since they can leverage diverse performance optimizations. For instance, CODE2INV can take advantage of GPUs and TPUs, and C2I can benefit from massive parallelization. Instead of comparing absolute running times, we observe that all four solvers are based on the Z3 theorem prover [6] and rely on the counterexamples from Z3 to adjust their search strategy. Therefore, we compare the number of queries to Z3, which is usually the performance bottleneck for verification tasks. We run all solvers on a single 2.4 GHz AMD CPU core up to 12 hours and using up to 4 GB memory for each program.

Figure 5a shows the number of instances solved by each solver and the corresponding number of queries to Z3. CODE2INV solves the largest number of instances, which is **106**. In contrast, ICE-DT, LOOPINVGEN and C2I solve **100**, **77** and **74** instances, respectively. ICE-DT heavily relies on predicate templates designed by human experts, which are insufficient for 19 instances that are successfully solved by CODE2INV. Furthermore, to solve the same amount of instances, CODE2INV costs orders of magnitude fewer queries to Z3 compared to the other solvers.

We also run CODE2INV using the time limit of one hour from the 2017 SyGuS competition. CODE2INV solves **92** instances within this time limit with the same hardware configuration. While it cannot outperform existing state-of-the-art solvers based on absolute running times, however, we believe its speed can be greatly improved by (1) pre-training on similar programs, which we show in Section 5.3; and (2) an optimized implementation that takes advantage of GPUs or TPUs.

CODE2INV is most related to C2I since both use accumulated counterexamples to adjust the sample distribution of loop invariants. The key difference is that C2I uses MCMC sampling whereas CODE2INV learns using RL. Figure 5b shows the sample complexity, i.e., number of candidates generated before successfully finding the desired loop invariant. We observe that CODE2INV needs orders of magnitude less samples which suggests that it is more efficient in learning from failures.

## 5.2 Ablation study

We next study the effectiveness of two key components in our framework via ablation experiments: counterexamples and attention mechanism. We use the same dataset as in Section 5.1. Table 1 shows our ablation study results. We see that besides providing a continuous reward, the use of counterexamples (CE) significantly reduces the verification cost, i.e., number of Z3 queries. On the other hand, the attention mechanism helps to reduce the training cost, i.e., number of parameter updates. Also, it

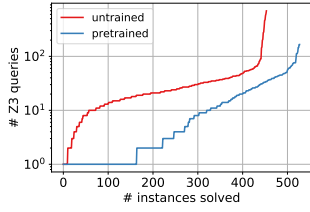

(a) with 1 confounding variable

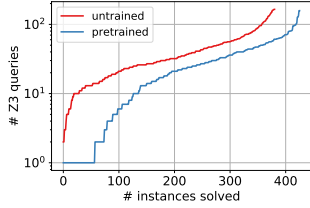

(b) with 5 confounding variables

```
int main()
{
    int a = 0, b = 0, c = 0;

    while(*) {
        a += 1;
        b += 1;
        c -= 1;
    }

    if(a != b) {
        assert (b == -1);
    }
    else{
    }
}
```

(c) attention for invariant a == b

```
int main()
{
    int n, k, c = 0;
    assume (n > 0);

    while(*) {
        if(c < n) {
            c = c + 1;
            k = 2;
        }
        if(c == n) {
            c = 1;
            k = 5;
        }
    }

    if(c == n) {
        assert (n > -1);
    }
}
```

(d) attention for the first part of invariant: c >= -1 && n >= 1

Figure 6: (a) and (b) are verification costs of pre-trained model and untrained model; (c) and (d) are attention highlights for two example programs.

helps to reduce the verification cost modestly. CODE2INV achieves the best performance with both components enabled—the configuration used in other parts of our evaluation.

Additionally, to test the effectiveness of neural graph embedding, we study a simpler encoding, that is, viewing a program as a sequence of tokens and encoding the sequence using an LSTM. The performance of this setup is shown in the last row of Table 1. With a simple LSTM embedding, CODE2INV solves 13 fewer instances and, moreover, requires significantly more parameter updates.

Table 1: Ablation study for different configurations of CODE2INV.

| configuration | #solved instances | max #Z3 queries | max #parameter updates |
|---|---|---|---|
| without CE, without attention | 91 | 415K | 441K |
| without CE, with attention | 94 | 147K | 162K |
| with CE, without attention | 95 | 392 | 337K |
| with CE, with attention | 106 | 276 | 290K |
| LSTM embedding + CE + attention | 93 | 32 | 661K |

## 5.3 Boosting the search with pre-training

We next address the question: given an agent that is pre-trained on programs $P_{train} = \{p_i\} \sim \mathcal{P}$, can the agent solve new programs $P_{test} = \{\tilde{p}_i\} \sim \mathcal{P}$ faster than solving from scratch? We prepare the training and testing data as follows. We take the programs solved by CODE2INV as the initial set and augment it by creating 100 variations for each of them by introducing confounding variables and statements in such a way that any valid loop invariant for the original program is still valid. Further details are provided in Appendix **??**. Finally, 90% of them serves as $P_{train}$, and the rest are used for $P_{test}$.

After pre-training the agent on $P_{train}$ for 50 epochs, we save the model and then reuse it for "fine tuning" (or active search [4]), *i.e.*, the agent continues the trial-and-error reinforcement learning, on $P_{test}$. Figure 6a and Figure 6b compare the verification costs between the pre-trained model and untrained model on datasets augmented with 1 and 5 confounding variables, respectively. We observe that, on one hand, the pre-trained model has a clear advantage over the untrained model on either dataset; but on the other hand, this gap reduces when more confounding variables are introduced. This result suggests an interesting future research direction: how to design a learning agent to effectively figure out loop invariant related variables from a potentially large number of confounding variables.

## 5.4 Attention visualization

Figure 6c and 6d show the attention highlights for two example programs. The original highlights are provided on the program graph representation described in Section 3.2.1. We manually converted the graphs back to source code for clarity. Figure 6c shows an interesting example for which CODE2INV

learns a strategy of showing the assertion is actually not reachable, and thus holds trivially. Figure 6d shows another interesting example for which CODE2INV performs a form of abductive reasoning.

## 5.5 Discussion of limitations

We conclude our study with a discussion of limitations. For most of the instances that CODE2INV fails to solve, we observe that the loop invariant can be expressed in a compact disjunctive normal form (DNF) representation, which is more suited for the decision tree learning approach with hand-crafted features. However, CODE2INV is designed to produce loop invariants in the conjunctive normal form (CNF). The reduction of loop invariants from DNF to CNF could incur an exponential blowup in size. An interesting future research direction concerns designing a learning agent that can flexibly switch between these two forms.

## 6 Related Work

We survey work in program synthesis, program learning, learning loop invariants, and learning combinatorial optimizations.

**Program synthesis.** Automatically synthesizing a program from its specification has been a key challenge problem since Manna and Waldinger's work [32]. In this context, syntax-guided synthesis (SyGuS) [9] was proposed as a common format to express these problems. Besides several implementations of SyGuS solvers [9, 33–35], a number of probabilistic techniques have been proposed to model syntactic aspects of programs and to accelerate synthesis[36–38]. While logical program synthesis approaches guarantee semantic correctness, they are chiefly limited by their scalability and requirement of rigorous specifications.

**Program learning.** There have been several attempts to learn general programs using neural networks. One large class of projects includes those attempting to use neural networks to accelerate the discovery of *conventional programs* [39–42]. Most existing works only consider specifications which are in the form input-output examples, where weak supervision [43–45] or more fine grained trace information is provided to help training. In our setting, there is no supervision for the ground truth loop invariant, and the agent needs to be able to compose a loop invariant purely from trial-and-error. Drawing inspiration from both programming languages and embedding methods, we build up an efficient learning agent that can perform end-to-end reasoning, in a way that mimics human experts.

**Learning program loop invariants.** Our work is closely related to recent work on learning loop invariants from either labeled ground truth [46] or active interactions with human experts [47]. Brockschmidt et al. [46] learn shape invariants for data structures (e.g. linked lists or trees). Their approach first extracts features using n-gram and reachability statistics over the program's heap graph and then applies supervised learning to train a neural network to map features to shape invariants. In contrast, we are concerned with general loop invariant generation, and our approach employs graph embedding directly on the program's AST and learns a generation policy without using ground truth as supervision. Bounov et al. [47] propose inferring loop invariants through gamification and crowdsourcing, which relieves the need for expertise in software verification, but still requires significant human effort. In contrast, an automated theorem prover suffices for our approach.

**Learning combinatorial optimizations.** Our work is also related to recent advances in combinatorial optimization using machine learning [4, 5, 48, 49]. However, as elaborated in Section 4, the problem we study is significantly more difficult, in the sense that the objective function is non-smooth (binary objective), and the positive reward is extremely sparse due to the exponentially growing size of the search space with respect to program size.

## 7 Conclusion

We studied the problem of learning loop invariants for program verification. Our proposed end-to-end reasoning framework learns to compose the solution automatically without any supervision. It solves a comparable number of benchmarks as the state-of-the-art solvers while requiring much fewer queries to a theorem prover. Moreover, after being pre-trained, it can generalize the strategy to new instances much faster than starting from scratch. In the future, we plan to extend the framework to discover loop invariants for larger programs which present more confounding variables, as well as to discover other kinds of program correctness properties such as *ranking functions* for proving program termination [50] and *separation predicates* for proving correctness of pointer-manipulating programs [51].

**Acknowledgments.** We thank the anonymous reviewers for insightful comments. We thank Ningning Xie for useful feedback. This research was supported in part by DARPA FA8750-15-2-0009, NSF (CCF-1526270, IIS-1350983, IIS-1639792, CNS-1704701) and ONR N00014-15-1-2340.

## Footnotes

[2]Our code and data are publicly available from https://github.com/PL-ML/code2inv

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
