[Supplementary Material · appendix.pdf]

# Appendix

## A   Details about learning

### A.1   Reward Design

Our reward function consists of two parts: *early reward* and *continuous reward*.

**Early reward**   The early reward constitutes quick feedback obtained by performing lightweight structure checks during the process of loop invariant generation. The goal is to quickly remove meaningless predicates that are trivially true (e.g. "e==e") or false (e.g. "e<e") or missing variables (e.g. "1<2"), or simple contradictions like "e1<e2 && e1>e2". Early reward is computed at the end of each action; if the partially generated invariant fails to pass the above checks, the generation process terminates immediately by returning a large negative reward -4; otherwise, a positive reward 0.5 is given. Note that one promising future work could be taking advantage of UNSAT cores from counterexamples to identify contradictory parts of the candidate invariant. These contradictory parts will be "non-trivial" contradictions, compared to trivial patterns we have considered.

**Continuous reward**   The goal of continuous reward is to reflect proof progress smoothly. It is computed after the loop invariant is generated and is based on three kinds of counterexamples. Let $ce_{pre}, ce_{inv}, ce_{post}$ denote the sets of counterexamples accumulated so far at the *pre*, *inv*, *post* step, respectively. Similarly, let $pass_{pre}, pass_{inv}, pass_{post}$ be the sets of counterexamples passed by current loop invariant candidate. The continuous reward is modeled as a function that takes these six sets of counterexamples as input and produces a scalar value. We used a simple but effective function, that is, the sum of ratios. Specifically, in the case no new counterexample is introduced, we used the sum of passed ratios of counterexamples:

$$\frac{|pass_{pre}|}{|ce_{pre}|} + \frac{|pass_{inv}|}{|ce_{inv}|} + \frac{|pass_{post}|}{|ce_{post}|}$$

When a new counterexample is returned, we used the staged sum:

$$\frac{|pass_{pre}|}{|ce_{pre}|} + [pass_{pre} = ce_{pre}]\frac{|pass_{inv}|}{|ce_{inv}|} + [pass_{pre} = ce_{pre}][pass_{inv} = ce_{inv}]\frac{|pass_{post}|}{|ce_{post}|}$$

where $[\cdot]$ is Iverson bracket. It examines counterexamples in an *ordered* way (i.e. *pre, inv, post*) so that counterexamples in the next step are considered only after all counterexamples in the previous step get passed. When we get the highest continuous reward, which is 3, we invoke the theorem prover to verify the current loop invariant candidate; if the theorem prover accepts it, then a correct loop invariant is found; otherwise, a new counterexample is returned, and we recompute the continuous reward according to the above reward function.

### A.2   Hyperparameters

By default, the embedding size used throughout the paper is 128. Batch size is set to 10. To compute the graph structured external memory representation, we run the message passing operator (as described in Equation (1)) for 20 steps. Learning rate is set to $0.001$ and fixed.

We maintain a circular buffer for the counterexamples. The buffer size is set to 100, i.e., we remove the old counterexamples when the buffer size is exceeded (although we seldom reached the 100 limit in our experiments). We use the counterexample to compute continuous feedback only after we collect 5 or more of them.

## B   Dataset

Our dataset is collected from recent literature [3, 7] and the 2017 SyGuS competition [31]. Dillig et al. [7] create a suite of 46 C programs for evaluation of loop invariant inference, on top of which Garg et al. [3] introduce 40 more benchmarks. The 2017 SyGuS competition consists of 74 benchmarks, which is in the SMT-LIB like format [52]. We manually convert benchmarks from SyGus competition to C programs, which have some overlaps with the above two benchmark suite, so we remove the overlapped ones. Figure 7 shows some example programs in the SyGuS challenge dataset.

```
1   int main() {
2       // variable declarations
3       int c;
4       int n;
5       // pre-conditions
6       (c = 0);
7       assume((n > 0));
8       // loop body
9       while (unknown()) {
10          {
11              if ( unknown() ) {
12                  if ( (c != n) )
13                  {
14                      (c  = (c + 1));
15                  }
16              } else {
17                  if ( (c == n) )
18                  {
19                      (c  = 1);
20                  }
21              }
22          }
23      }
24
25      }
26      // post-condition
27  if ( (c == n) )
28  assert( (n > -1) );
29
30  }
```

```
1   int main() {
2       // variable declarations
3       int x;
4       int y;
5       // pre-conditions
6       (x = 1);
7       (y = 0);
8       // loop body
9       while ((y < 100000)) {
10          {
11              (x   = (x + y));
12              (y   = (y + 1));
13          }
14
15      }
16      // post-condition
17      assert( (x >= y) );
18  }
```

```
1   int main() {
2       // variable declarations
3       int c;
4       int y;
5       int z;
6       // pre-conditions
7       (c = 0);
8       assume((y >= 0));
9       assume((y >= 127));
10      (z = (36 * y));
11      // loop body
12      while (unknown()) {
13          if ( (c < 36) )
14          {
15              (z   = (z + 1));
16              (c   = (c + 1));
17          }
18
19      }
20      // post-condition
21  if ( (c < 36) )
22  assert( (z < 4608) );
23  }
```

```
1   int main() {
2       // variable declarations
3       int x;
4       int y;
5       int z1;
6       int z2;
7       int z3;
8       // pre-conditions
9       assume((x >= 0));
10      assume((x <= 2));
11      assume((y <= 2));
12      assume((y >= 0));
13      // loop body
14      while (unknown()) {
15          {
16              (x   = (x + 2));
17              (y   = (y + 2));
18          }
19
20      }
21      // post-condition
22  if ( (x == 4) )
23  assert( (y != 0) );
24
25  }
```

Figure 7: Examples of programs in SyGuS challenge dataset (after converting to C).

### B.1 Training data augmentation

We augment the training dataset on top of the set of programs that CODE2INV can solve. Given a program, we first randomly select an integer K ranging from 1 to 5, which is the number of confounding variables to insert. Then, we initialize each newly created variable with a value ranging from -100 and 100. After that, we insert a statement after each statement in the loop body. The newly inserted statement only uses confounding variables and constants so that any valid loop invariant in the original program is still valid after augmentation. The *lvalue* of the statement is randomly and uniformly sampled from confounding variables, and for the *rvalue* expression of statement, we first randomly choose a depth (either 1 or 2) for the AST tree of the expression, then randomly and uniformly pick an operator from $\{ +, -, * \}$, and operands from confounding variables and constants ranging between -100 and 100. For each program and each chosen parameter K, we repeat the above process a 100 times.

## C   Attention score interpretation

The learned attention scores measure the importance of different nodes in the program graph memory. In Figure 6c and 6d we converted the attention into the raw program file. Here we include the original graphs, together with the attentions that represented by different scale of colors. See Figure 8 and 9.

Figure 8: Attention over program graph, which converts to Figure 6c in main text.

Figure 9: Attention over program graph, which converts to Figure 6d in main text.