[Reviews · NeurIPS 2018]

Reviewer 1



The paper presents a novel deep network architecture termed DELPHI to automatically infer loop invariants for use in program verification. The architecture takes in as input source code which has (1) a number of assumption or assignment statements, (2) a loop with nested if-else statements with arithmetic operations and (3) a final assertion statement. The output of the architecture is a loop invariant in CNF which holds true at every iteration in the loop, and for which the assertion (3) is true after the loop ends execution. The architecture represents the source code AST using a graph-structured neural network, and treats it as a structured memory which it repeatedly accesses through attention operations. The generation of the CNF invariant is broken up into a sequential decision-making process where at each time the architecture predicts an output (op, T), where op is either && or || and T is a simple logical expression. The partial CNF invariant is re-encoded by a Tree LSTM after each action output, and the next step prediction is conditioned on this information. Due to the lack of available ground-truth labels, the entire system is trained using RL (the loop invariant is either validated or rejected by an automated theorem prover, Z3). The reinforcement signal is the binary success/failure returned by Z3 along with some shaping rewards, with some based on the types of counterexamples that are returned by Z3. The architecture is empirically validated on a set of 133 benchmark programs, and is compared against previous state-of-the-art solvers which span a variety of methods from stochastic search to decision-tree-based heuristics. Results show that DELPHI is highly competitive with the state-of-the-art and, despite not beating every previous method, requires fewer queries to Z3. Additionally, DELPHI is shown to demonstrate the ability to transfer knowledge, pre-training the architecture on a set of related programs provides a significant improvement in performance when learning is continued on a “test” set. The paper’s writing is clear and the description of the architecture and the methods used to train it is well explained. Additionally, the motivation behind DELPHI as perhaps imitating the human process behind loop invariant generation provides intuition behind the architectural choices. The paper presents a step towards automatic program verification using deep neural networks. DELPHI’s performance is highly competitive with the current state-of-the-art, and this is especially promising given that (1) DELPHI demonstrates transfer if pretrained on a set of programs and (2) the currently available loop invariant datasets are very small (on the order of 133 programs). Given that deep networks often work better the more data you have, this suggests that potentially DELPHI’s performance could improve even further if more data was available. The architecture is clearly defined, and detailed explanations of the parts of the network are provided. The use of attention over the program AST encoded as a graph-neural-network was interesting. An ablation experiment suggested that each component of the model provided some benefit, and that not one part was solely responsible for the success of the model. A potential weakness is that the metric used for comparison between baselines, number of queries to the Z3 theorem prover, is not necessarily a practical performance measure. It is not clear how the methods compare to each other in terms of absolute wall clock time or some measure of compute. It is understandable that DELPHI could do worse in these measures (with the consideration that future work could potentially narrow the gap), but I think it would still be interesting to see what the current difference is. Additionally, some architectural choices were made without necessarily testing whether simpler baselines worked better (e.g. would attention over the linearized program encoded by an LSTM work just as well as the program encoded by the graph-structured neural network?). Finally, the related work is only a short section in the appendix, which I believe can be expanded on and should be pushed into the main paper. In summary, I believe this work presents an interesting step towards deep program verification. The experiments are relatively comprehensive, the architecture is interesting and intuitive, and the results demonstrate not only the feasibility but the state-of-the-art competitiveness of combining deep architectures and RL in the context of verification. It also provides some evidence that deep architectures are conducive to transfer, meaning that their performance will likely improve with the size of the datasets available, in contrast to the purely search-based methods.

Reviewer 2



= Summary The paper presents a new method for reinforcement learning-based counterexample-guided loop invariant generation. First the program's source code is encoded into a vector representation by transforming it into a graph and applying a graph neural network. Then, a tree decoder is used to generate an arithmetic formula in conjunctive normal form, using attention over the encoded program. Overall, the approach seems promising, and the shown experimental results indicate that it is an interesting alternative to standard deductive and more recently introduced data-driven CEGAR approaches to program verification. However, the paper does not actually show the core of its contribution, the smoothed reward function used to train the system, and so I believe that the submission in its current form can not be accepted. I am inviting the authors to clarify this point (cf. below for details) in the rebuttal and update the paper subsequently, and would be happy to increase my score then. [UPDATE: This happened.] = Quality The paper follows recent trends in the literature (programs as graphs, syntax-directed decoding of formulas, ...), but combines it with a novel reinforcement learning approach, sidestepping the issue of a lack of labeled examples and making it easier to adapt the method to new domains. However, the core of the main contribution, namely the reward function that smoothes over the very sparse, binary valid/invalid signal, is not defined in the main paper. Instead, it refers to the supplement, where the reward is again not defined. Instead, some handwavy text about the use of counterexamples is provided (for example, do their values / traces matter, or just their number? Are UNSAT cores used to identify which parts of the candidate invariant are contradictory?). Without this, the contribution is not remotely reproducible and the paper cannot be accepted. The main text of the paper has to contain at least an approximate definition of the reward function, and the supplement should provide a full definition. Documenting the use of counterexamples is especially important, as Sect. 5 even provides an ablation study underlining their importance. [UPDATE: The authors have provided details on the reward function in their response, and promised to include these details in a revision of their paper. Based on this, I think an interesting question for a future revision would be to experimentally valid the design of the reward function. For example, inverting the order of reward components from pre->inv->post to post->inv->pre would be interesting (corresponding to a backwards-looking style, as in Hoare logic proofs); similarly not ordering the components at all could be an interesting ablation.] Similarly, the paper should be more honest about the runtime performance of the proposed approach. Lines 260-265 introduce the notion of calls to the backing SMT solver as a metric; which is fine to show the data efficiency of the approach, but wallclock time is interesting for actual use of an invariant generation tool in a verifier. The timeout of 12 hours used in the paper contrasts with 1 hour allowed in the SyGuS competition from which many benchmarks are taken, and the fact that the ICE-DT solver requires less than 5 _seconds_ for most benchmarks (cf. https://dl.acm.org/citation.cfm?doid=2837614.2837664). Even if the method is currently not competitive in terms of runtime, this should be stated explicitly in the paper. [UPDATE: The authors have promised to include runtime in the next revision.] = Clarity The paper succeeds reasonably well in introducing basic verification concepts in the little space it has, and apart from the lack of description of the reward function is reasonably clear. I found Sect. 3.1 not particularly enlightening (especially because I don't see that it supports conclusions (1) and (2) in lines 128-131) and would rather use some of the space used for the example to go into more detail on the reward function, which is the actual contribution of the paper. = Originality Finding a smoothing of the reward function that actually works is a novel and interesting contribution that hasn't been provided before. The authors review most of the related literature and discuss how they are clearly differing from it, though they seem to be unaware of one related work that infers loop invariants for heap programs using a tree decoder and a graph encoding of its inputs (but uses fully supervised training on synthetic data): Marc Brockschmidt, Yuxin Chen, Pushmeet Kohli, Siddharth Krishna, and Daniel Tarlow. Learning Shape Analysis. In SAS'17. = Significance Pushing beyond existing heuristics in deductive invariant generation tools is of great importance to making program verification more generally useful; several large-scale efforts (e.g., the verified TLS stack, the seL4 kernel, ...) towards formally verified software could profit from an evolution of the presented approach to more interesting domains. [This is minor pet peeve that I have with the paper: The fact that integer invariants are interesting, but not all that useful without heap invariants is never stated] = Minor notes - lines 31/32: Calling Octagons "hand-crafted features" seems unnecessarily harsh. It would be more correct to call them a restricted hypothesis space. - line 74: This statement is just wrong, for example for arbitrary polynomials or for the very relevant domain of separation logic. It's true for the domain of linear integer arithmetic that is investigated in the paper, but that distinction isn't stated. - line 141: "... module summarizes what have been" - s/have/has/ - line 159: "To convert the graph into vector representation" - missing article.

Reviewer 3



The author propose an end-to-end reasoning framework which can compose solutions without supervision. They present highly convincing experiments where they can demonstrate that their approach solves a comparable number of benchmark problems as a state-of-the-art theorem prover but with much fewer queries. I have background in inductive programming but nor background in theorem proving and in end-to-end learning. So maybe my enthusiasm is due to some lack in knowledge. In my opinion, the presented approach has a high quality and the results are original and significant. The paper is carefully written and a supplement provides details about the dataset and the competition.